# Vibration Energy Harvesting from the Subwavelength Interface State of a Topological Metamaterial Beam

**DOI:** 10.3390/mi13060862

**Published:** 2022-05-30

**Authors:** Yongling Lu, Zhen Wang, Xueqiong Zhu, Chengbo Hu, Jinggang Yang, Yipeng Wu

**Affiliations:** 1Research Institute of State Grid Jiangsu Electric Power Co., Ltd., Nanjing 211103, China; 15105182955@163.com (Y.L.); wangzhenscut@163.com (Z.W.); 18795897606@163.com (X.Z.); 15105168989@163.com (C.H.); 15105168828@163.com (J.Y.); 2State Key Laboratory of Mechanics and Control of Mechanical Structures, Nanjing University of Aeronautics and Astronautics, Nanjing 210016, China

**Keywords:** piezoelectric energy harvesting, topological metamaterial, interface state, defect

## Abstract

Topological metamaterial has been a research hotpot in both physics and engineering due to its unique ability of wave manipulation. The topological interface state, which can efficiently and robustly centralize the elastic wave energy, is promising to attain high-performance energy harvesting. Since most of environmental vibration energy is in low frequency range, the interface state is required to be designed at subwavelength range. To this end, this paper developed a topological metamaterial beam with local resonators and studied its energy-harvesting performance. First, the unit cell of this topological metamaterial beam consists of a host beam with two pairs of parasitic beams with tip mass. Then, the band structure and topological features are determined. It is revealed that by tuning the distance between these two pairs of parasitic beams, band inversion where topological features inverse can be obtained. Then, two sub-chains, their design based on two topologically distinct unit cells, are assembled together with a piezoelectric transducer placed at the conjunction, yielding the locally resonant, topological, metamaterial, beam-based piezoelectric energy harvester. After that, its transmittance property and output power were obtained by using the frequency domain analysis of COMSOL Multiphysics. It is clear that the subwavelength interface state is obtained at the band-folding bandgap. Meanwhile, in the interface state, elastic wave energy is successfully centralized at the conjunction. From the response distribution, it is found that the maximum response takes place on the parasitic beam rather than the host beam. Therefore, the piezoelectric transducer is recommended to be placed on the parasitic beam rather than host beam. Finally, the robustness of the topological interface state and its potential advantages on energy harvesting were studied by introducing a local defect. It is clear that in the interface state, the maximum response is always located at the conjunction regardless of the defect degree and location. In other words, the piezoelectric transducer placed at the conjunction can maintain a stable and high-efficiency output power in the interface state, which makes the whole system very reliable in practical implementation.

## 1. Introduction

During the past few decades, micro-scale electronics have been developed quite fast. More and more low-power consumption devices have been used in the area of wireless sensors, portable devices, structural health monitoring and the internet-of-things. The piezoelectric energy harvester has become a promising way of capturing wasted vibration energy in the environment to provide the sufficient power for small devices [1,2,3].

To sufficiently harvest vibration energy from the environment, researchers have devoted a lot of effort to energy-harvesting structures and circuits. The linear piezoelectric energy harvester [4] is very stable in generating great power when it resonates. However, the bandwidth of resonance of a linear system is generally narrow [5]. To enlarge the operation bandwidth, there are mainly two categories of methods. The first one is an adaptive energy harvester [6,7,8,9]. The key to the adaptive energy harvester is matching the natural frequency of the energy harvesting structure with the frequency of vibration so that the system can exhibit resonance and generate large output power. The second one is nonlinear energy harvesting, which mainly uses the unique features of nonlinear systems to attain bandwidth enlargement. For example, based on the duffing oscillator, monostable vibration energy harvesters (VEHs) [10,11,12,13], bistable VEHs [14,15,16], tristable VEHs [17,18,19] and multistable VEHs [20,21] have been developed. The main advantage of the duffing-type nonlinear VEHs is that the high energy oscillation can exhibit over a very wide band. More recently, it was also found that the multi-degree-of-freedom nonlinear VEHs, such as internal resonance based VEHs [22,23] and magnetically coupled VEHs [24] can further enlarge the operation bandwidth thanks to the multi orbits of high-energy oscillation. However, those two methods have their own limitations. For the adaptive energy harvester, the way of tuning natural frequency usually requires external force or power, which inversely reduces the output power [8]. For the nonlinear energy harvester, the high energy oscillation is sensitive to the external perturbation, resulting in an easy jump from a high-energy orbit into a low-energy orbit [25,26].

Recently, metamaterial has been introduced into vibration energy harvesting owing to its unique features of wave manipulation. For example, Carrara et al. [27] developed a defect-metamaterial-based energy harvester by using the defect mode to centralize wave energy and improve energy-harvesting efficiency. Tol et al. [28] developed a gradient-index phononic crystal lens VEH to attain large power over a wide band by utilizing wave-focusing features. A locally resonant metamaterial VEH was developed by Gonella et al. [29] to attain concurrent energy harvesting and vibration suppression. It was found that vibration energy can be suppressed inside the bandgap and harvested outside the bandgap. To further improve energy harvesting, Hu et al. [30,31] proposed an internally coupled locally resonant metamaterial VEH and attained four times more power than that of the conventional meta-VEH. However, one critical problem is free to be explored. The wave manipulation capability of these aforementioned metamaterials is sensitive to additional defects induced by material fatigue or external interferences, especially for the defect mode and Lens-type metamaterial, where local defect may largely reduce the effectiveness of wave focusing. Therefore, how to improve the robustness of metamaterial- based VEH is a critical problem to be solved in practical implementation.

More recently, topological metamaterial, developed from the topological insulator in condensed matter, has shown great advantages on robust wave guiding and manipulation, owing to topological protection features [32,33,34]. For example, Wang et al. [32] found the Fano resonance in a topological metamaterial is robust against random perturbations. Therefore, topological metamaterial is very promising to maintain robust energy harvesting. Fan et al. [35] proposed an acoustic energy harvester based on a one-dimensional phononic crystal tube. Lan et al. [36] theoretically studied the potential advantage of a topological-metamaterial-based vibration energy harvester based on the mass-spring mode. Ma et al. [37] conducted experimental tests on the energy-harvesting performance of a topological phononic crystal beam. Wen et al. [38] proposed a topological phononic plate for robust energy harvesting. All these investigations have shown the topological metamaterial can attain high output power in the interface state. 

However, the main limitation of topological-phononic-crystal based vibration energy harvesters is the frequency of the interface state of the phononic crystal (PC)-type VEHs, which is relatively higher than the frequency of most environmental vibrations. A feasible way to deal with this frequency mismatching problem is by utilizing the subwavelength interface state for low-frequency vibration-energy harvesting. To this end, this paper proposed a topological metamaterial beam VEH with local resonators to attain the subwavelength interface and study its energy-harvesting performance, including power output and robustness. The main contents of this paper are as follows: Section 2 gives the design process of the unit cell and the topological metamaterial beam VEH. The topological property and band inversion features are studied. Section 3 studies the subwavelength interface state and energy-harvesting performance, following the robustness analysis of a defected interface state in Section 4. Several meaningful conclusions are drawn in Section 5.

## 2. Design of a Locally Resonant Metamaterial Beam

### 2.1. Mass-Spring Model

To design a 1D topological metamaterial, the Su-Schrieffer-Heeger (SSH) model [39] has been widely used. For example, a topological PC beam was proposed by Yin et al. [40] based on the SSH model, and the topological edge state was clearly obtained in the experiment. A topological PC rod was studied by Muhammad et al. [41], which is based on the SSH model as well. To obtain the subwavelength topological interface state, the local resonator was introduced into the SSH model, yielding a topological locally resonant metamaterial. The topological interface state in a locally resonant acoustic system was theoretically obtained by Zhao et al. [42]. It was found that the interface state in the locally resonant metamaterial takes place in the band-folding bandgap, which is lower than the locally resonant bandgap. Then, Fan et al. [35]. found that the interface state can be tuned by adjusting the local resonator, which indicates that the interface state can be obtained in the subwavelength range in a topological locally resonant metamaterial. 

In the SSH model, the metamaterial consists of two topologically distinct sub-chains and the topological interface state take place at the conjunction of these two sub-chains. Therefore, in the design of topological metamaterial, the first step is finding two sub-chains with different topology properties. For one dimensional chain, the topology property can be determined by the topological invariant Zak phase. Figure 1a shows the unit cell of the mass-spring model of a topological locally resonant lattice. The unit cell consists of a diatomic chain with mass-in-mass local resonator. In the theoretical study [42], it is found that the Zak phase is 0 when *k*_1_ > *k*_2_, whereas it is π when *k*_1_ < *k*_2_. Therefore, we can tune the Zak phase of the unit cell by exchanging the stiffnesses of outer mass (*k*_1_ and *k*_2_, *k*_1_ ≠ *k*_2_). Besides, since the Zak phase is irrelevant with the local resonator, the parameters of the local resonator in each cell are set to be same. Hence, we can have two different configurations of unit cells with different topological features. After that, the second step is fabricating two topologically distinct sub-chains with these two unit-cells, respectively. Finally, the topological metamaterial is obtained by assembling these two sub-chains together. Figure 1b depicts the infinitely long system of the mass-spring model of a topological locally resonant metamaterial.

### 2.2. Beam Model

To utilize the subwavelength interface state, we developed a topological locally resonant metamaterial beam based on the mass-spring model. Figure 2a is the unit cell of the proposed topological metamaterial beam. It consists of a host beam and two pairs of parasitic beams. These parasitic beams with a tip mass can be treated as a local resonator. The resonance frequency of these parasitic beams can be tuned by adjusting the size of the tip mass. The reason why we use a pair of parasitic beams is to make sure that the parasitic beam and the host beam exhibit the bending mode at the same time. All these parasitic beams share the same size, whereas their locations on the host beam are tuned to obtain different topological features of unit cell. Figure 2b,c gives the side view and front view of the unit cell. The length of the unit cell is *L* and the distance of these two pairs of parasitic beams is *L_d_*. Table 1 lists the parameters of the unit cell. Notably, in the design of unit cell, the parameter *L_d_* is tuned to obtain different topology features whereas other parameters are kept constant.

Then, the dispersion relation of the proposed unit cell is studied by using COMSOL Multiphysics. Figure 3 gives the dispersion relation of unit cells with different *L_d_*. For convenience, configurations *C*_1_, *C*_2_ and *C*_3_ refer to the unit cells with *L_d_* = 3 mm, *L_d_* = 8 mm and *L_d_* = 13 mm, respectively. It is found that when *L_d_* is 3 mm (Figure 3a), there are two bandgaps. From the mode of unit cell, it is learned that the upper bandgap induced by the local resonators is a locally resonant bandgap. The lower one is band folding induced bandgap, which is also a Bragg scattering (BS) bandgap. This BS bandgap is induced by the impedance mismatch of the left and right parts of the unit cell. When *L_d_* is tuned to be 8 mm (Figure 3b), only one bandgap (LR bandgap) is observed. The main reason for the close of the BS bandgap is that when *L_d_* = 8 mm, the left and right parts of the unit cell are the same and the impedance mismatch between these two parts disappears. When *L_d_* is tuned to 13 mm (Figure 3c), it is found that the closed BS bandgap opens again, and the band structure of *L_d_* = 13 mm is very similar to that of *L_d_* = 3 mm. Then, the topology feature of the unit cell of the one-dimensional lattice is determined by the Zak phase of dispersion relation [43], which can be calculated as follows:(1)θnZak=−Im∑i=1Nln[12ρv2∫unitcell[un,ki*(x,r)un,ki+1(x,r)]drdx]
where *x* is the axial coordinate, *r* is the positions in the cross-sectional plane, *ρ* = 1.3 kgm^−3^ is the air density, *v* = 343 m/s is the speed of sound in the air, *N* is the point number that we selected from *k* = π/*L* to *k* = −π/*L* and *u_n_*_,*k*_(*x*,*r*) is the periodic in-cell part of the normalized Bloch pressure eigenfunction of a state in the nth band with wave vector *k*. The detail process of calculating Zak phase follows the method proposed in reference [43].

It is found that the Zak phase of the BS bandgap (*θ*_Zak_) of configuration *C*_1_(*L_d_* = 3 mm) is π, whereas that of configuration *C*_3_ (*L_d_* = 13 mm) is 0, which indicates that configurations *C*_1_ and *C*_3_ are topologically distinct. To further study the topology features of these two configurations, the effect of *L_d_* on the band edge of the BS bandgap and the corresponding eigenmodes is studied. Figure 4 shows that when *L_d_* increases from 2 mm to 14 mm, the bandwidth of the BS bandgap starts to decrease at the beginning and becomes zero at 8 mm; after that, the BS bandgap reopens and the bandwidth increases as well. From the eigenmodes, it is found that for configuration *C*_1_, the eigenmode at the upper edge is asymmetrical with respect to the central cross-sectional plane of the unit cell, whereas the eigenmode at the lower edge is symmetrical with respect to the central cross-sectional plane. For configuration *C*_3_, it is exactly the opposite. The upper edge is symmetrical whereas the lower edge is asymmetrical. This implies that the symmetry of band edge states can be reversed by tuning *L_d_*. Such a feature is called a band inversion, which is an analogue to the band inversion process in quantum physics. 

Based on the Zak phase analysis and the band inversion property, a topological metamaterial can be designed based on these two unit cells. Figure 5 describes the formation of a topological locally resonant metamaterial beam. At first, the unit cells *C*_1_ and *C*_3_ are used to construct two different sub-chains. Then, these two sub-chains are connected to obtain the topological metamaterial beam.

## 3. Subwavelength Interface State and Energy Harvesting

To study the dynamics and energy harvesting performance of the proposed system, numerical simulations on the transmittance and output power are conducted. The cell numbers of left and right sub-metamaterials are five. The whole system is designed to be a cantilever metamaterial beam. The left boundary is a fixed end whereas the right one is a free end. The material of the beam is resin, which is widely used in 3D printing. The piezoelectric transducer placed at the conjunction is connected with a pure resistance load *R*. The size of the piezoelectric transducer (PZT-5H) is 5 mm × 8 mm × 1 mm. The whole system is driven by base acceleration excitation. The excitation point is the left fixed end whereas the detection point is the right free end. The transmittance of this topological metamaterial is simulated by the frequency domain analysis of COMSOL Multiphysics. The algorithm used is Multifrontal Massively Parallel sparse direct solver (MUMPS), a default method in the frequency domain analysis of COMSOL Multiphysics.

Figure 6 shows the transmittance and strain distribution of this topological metamaterial VEH. Two bandgaps are clearly obtained. The first bandgap is from 565 Hz to 678 Hz, which is the band-folding-induced BS bandgap. The second bandgap is from 713 Hz to 987 Hz, which is the locally resonant bandgap induced by the parasitic beams. The topological interface state takes place at 607 Hz. From the displacement distribution, it is learned that the elastic wave can be quickly suppressed in both BS bandgap and LR bandgap (Figure 6b,d). However, in the interface state (Figure 6c), the elastic wave is successfully centralized at the conjunction, resulting in a large amplitude response. Therefore, a piezoelectric transducer is placed at the conjunction to maximize the output power. From the response distribution, it is learned that the parasitic beam owes a larger deformation than the host beam. Therefore, the piezoelectric transducer is mounted on the parasitic beam rather than the host beam. Then, it is found that large output power is obtained at the interface state, as shown in Figure 7a. By tuning the load resistance, the output power can be maximized, and the peak power reaches 214.2 μW at 200 kΩ (Figure 7b). Therefore, we can conclude that the interface state can be used to efficiently harvest the low frequency vibration energy by a topological locally resonant metamaterial beam.

## 4. Robustness to Local Defect

Since the most attractive feature of the interface state is topological protection, it is reasonable to study the effect of the local defect on the dynamics and energy-harvesting performance of the subwavelength interface state. The local defect is introduced by decreasing the thickness of the beam at the *n*th cell. The length of defect is set to be *S_d_* = 2.4 mm, whereas the thickness and location of the defect is varied to evaluate the effect of the defect. First, the effect of the defect’s thickness is studied. The defect is assumed to be located at the conjunction of the topological metamaterial beam, as shown in Figure 8a. The thicknesses of defects are set to be 0 mm, 0.25 mm, 0.5 mm, 0.75 mm and 1 mm, respectively. Then, the output voltages of the interface states of these defected topological metamaterial beams are obtained and shown in Figure 8d. It is clearly found that as the defect’s thickness increases, the interface state will shift to the lower frequency, whereas the output voltage will increase gradually. The potential reason for the decreasing resonance frequency is due to the reduced stiffness at the conjunction. Figure 8b,c compares the displacement distribution of the interface state of the topological metamaterial beam with/without defected conjunction. It is revealed that both of them have the maximum response at the conjunction position, which indicates that the wave energy is successfully localized at the conjunction by both the perfect and defected interface states. From the perspective of energy harvesting, introducing a defect at the conjunction can help the interface state by concentrating more energy at the interface, which is beneficial to improve the output voltage.

Then, the effect of defect position on the dynamics and performance is evaluated. In this case, the position of defect is set to be the 5th, 6th, 7th, 8th and 9th cell, respectively, whereas the thickness and length of the defect are kept constant with *h_d_* = 1 mm, *S_d_* = 2.4 mm. Figure 9a gives the frequency–voltage relations of these topological metamaterial beams with the defect located at different positions. It is clearly found that as the defect moves away from the conjunction, the resonance peak of interface state tends to approach the interface state of a perfect topological beam without defect, and the output voltage decreases as well. From the perspective of energy harvesting, the defect is preferred to be arranged at the conjunction to maximize the energy-harvesting efficiency of the interface state. Moreover, it is interestingly found in Figure 9b that although the defect locates at different places, the maximum responses of these different defected interface states all take place at the conjunction, which indicates that the wave localization ability of the interface state is insensitive to the local defect, showing strong robustness towards the local defect. In practical application, local defects induced by material fatigue, manufacture precision and external interferences are very common and can be harmful by significantly reducing the energy-harvesting efficiency. By using the topological metamaterial, the high-energy-harvesting performance can be guaranteed since the elastic wave energy can always be centralized to the place where the piezoelectric transducer is located.

## 5. Results

Robustness is one of the most critical problems for energy harvesters based on wave focusing/localization. Topological interface state is a novel and emerging method to focus wave with high robustness, owing to its topological protection feature. Therefore, topological metamaterial-based VEH can be a potential solution to robust energy harvesting. Meanwhile, the topological interface state is preferred to be designed at the subwavelength range since the frequency of the environmental vibration is relatively low. To this end, this paper introduced the locally resonant topological metamaterial into low frequency vibration energy harvesting. A topological metamaterial beam with local resonators was proposed based on the mass-spring model and the topological features were determined. Then, the dynamics and energy harvesting performance of subwavelength interface state was studied. Moreover, a local defect was introduced into the proposed metamaterial VEH to evaluate the robustness. From the analysis and comparison, several useful conclusions are obtained: (1)The subwavelength topological interface state can effectively localize the elastic wave energy at the conjunction, resulting in a significant improvement of output power.(2)Since the maximum deformation takes place at the local resonator rather than host beam, the piezoelectric transducer is recommended to be bonded at the parasitic beam.(3)The wave energy is always localized at the conjunction in the interface state regardless of the location and degree of local defect, which indicates that the proposed topological metamaterial beam owns a very good robustness to local defect and can be a promising solution to achieve robust vibration energy harvesting in practical implementation.

## Figures and Tables

**Figure 1 micromachines-13-00862-f001:**
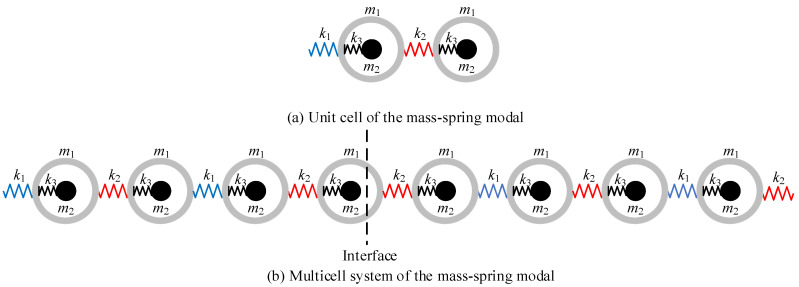
Mass-spring model and beam model of a locally resonant topological metamaterial. (**a**) Unit cell of the mass-spring model; (**b**) Multicell system of the mass-spring model.

**Figure 2 micromachines-13-00862-f002:**
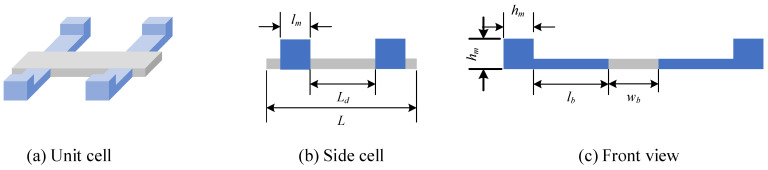
Unit cell of a locally resonant topological metamaterial beam.

**Figure 3 micromachines-13-00862-f003:**
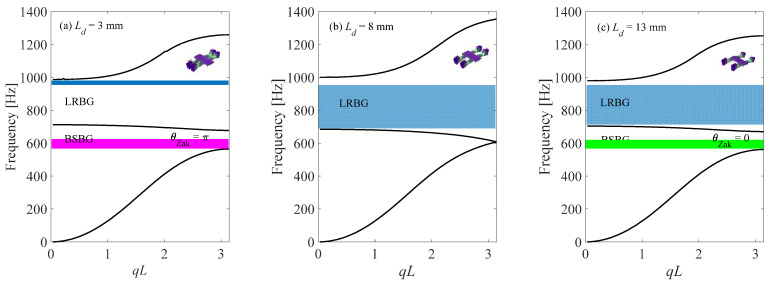
Band structures and Zak phase of a locally resonant topological metamaterial beams for different *L_d_*: (**a**) *L_d_* = 3 mm, (**b**) *L_d_* = 8 mm, (**c**) *L_d_* = 13 mm.

**Figure 4 micromachines-13-00862-f004:**
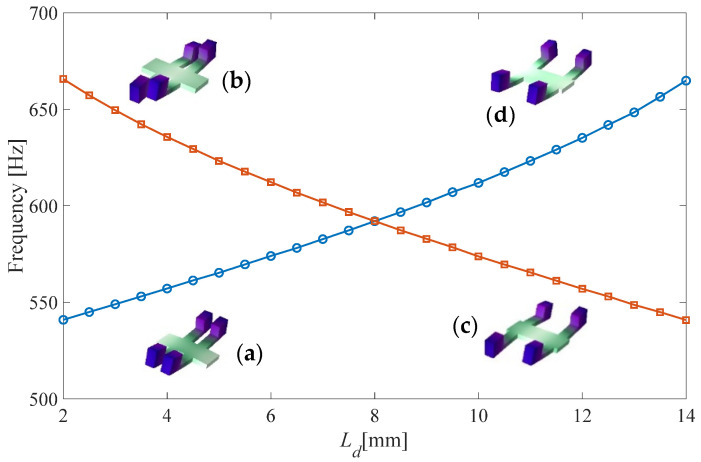
Eigenfrequencies and eigenmodes of the band edges as a function of *L_d_*: (**a**) first eigenmode at *L_d_* = 3 mm, (**b**) second eigenmode at *L_d_* = 3 mm, (**c**) first eigenmode at *L_d_* = 13 mm, (**d**) second eigenmode at *L_d_* = 13 mm.

**Figure 5 micromachines-13-00862-f005:**
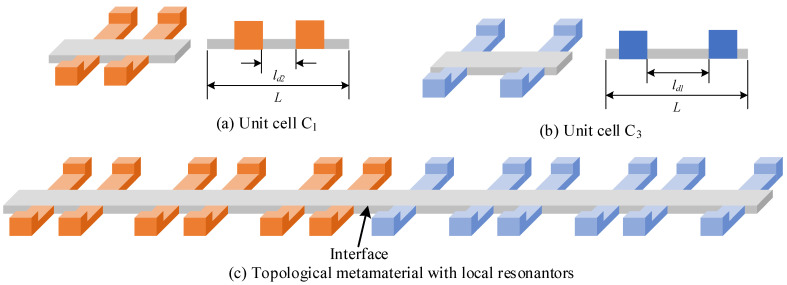
Mass-spring model and beam model of a locally resonant topological metamaterial: (**a**) and (**b**) are two different unit cells with different topology features, and (**c**) is the topological locally resonant metamaterial beam.

**Figure 6 micromachines-13-00862-f006:**
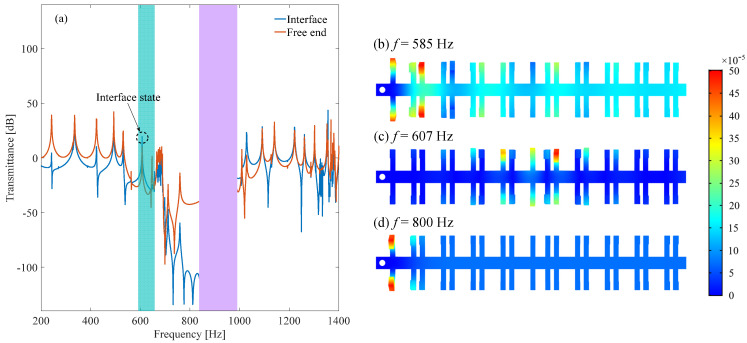
Transmittance and response distribution of the proposed topological metamaterial beam: (**a**) is the transmittances at the interface and free end; (**b**–**d**) are the response distributions at different excitation frequency.

**Figure 7 micromachines-13-00862-f007:**
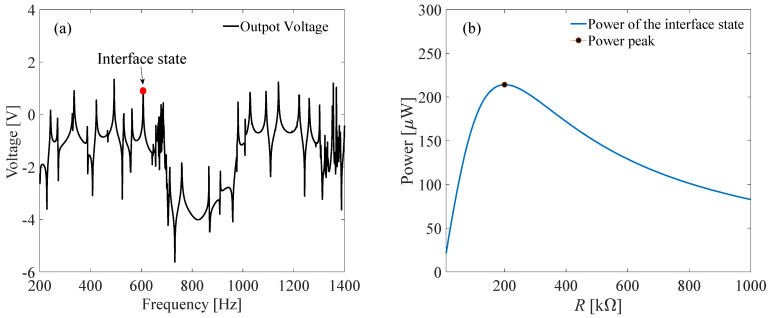
Voltage and optimal power of the proposed topological metamaterial beam: (**a**) Frequency-voltage response; (**b**) resistance versus power.

**Figure 8 micromachines-13-00862-f008:**
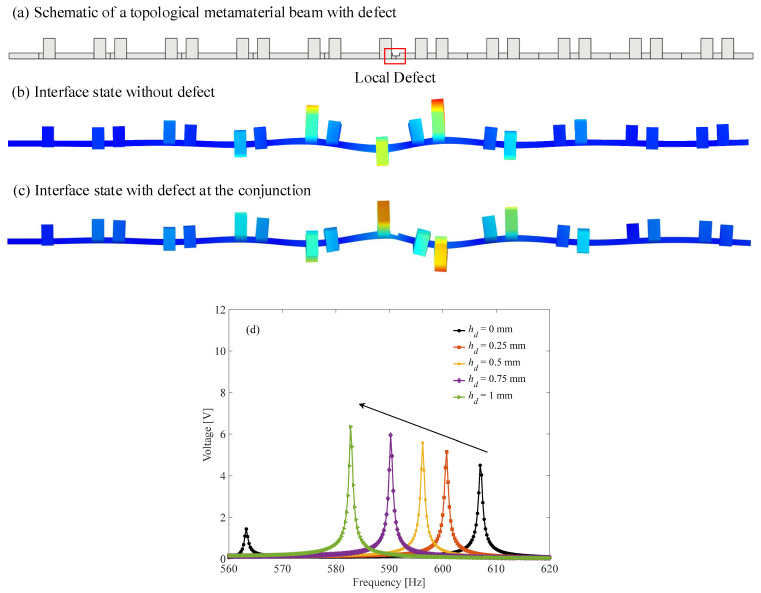
Effect of defect thickness on the interface state and output voltage (*A* = 10 g, *R* = 1 MΩ): (**a**) schematic of a topological metamaterial beam with defect; (**b**) interface state without defect; (**c**) interface state with defect at the conjunction; (**d**) the output voltages of the interface states.

**Figure 9 micromachines-13-00862-f009:**
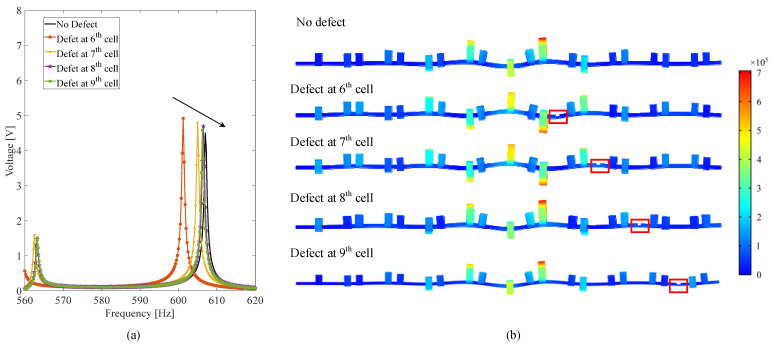
Effect of defect location on the interface state and output voltage: (**a**) the frequency–voltage relations of the topological metamaterial beam; (**b**) interface states at various conditions.

**Table 1 micromachines-13-00862-t001:** Parameters of the unit cell.

Parameters	Value	Parameters	Value
Length of unit cell, *L*	24 mm	Width/Hight of tip mass, *h_m_*	7 mm
Width of host beam, *b*	10 mm	Distance between two parasitic beams, *l_d_*	4 mm
Height of main/parasitic beam, *h*	2 mm	Material density, *ρ_m_*	1180 kg/m^3^
Length of parasitic beam, *l_b_*	10 mm	Poisson’s ratio, *μ*	0.3
Length of tip mass, *l_m_*	4 mm	Yang’s Elastic Modulus, E	2.5 Mpa

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
