# Peer review of "Vibration Energy Harvesting from the Subwavelength Interface State of a Topological Metamaterial Beam"

_micromachines, 2022, doi:10.3390/mi13060862_

Round 1
Reviewer 1 Report
This manuscript proposed a new metamaterial beam consisting of local resonators, which can obtain vibration energy in the relatively low-frequency domain. Element analysis and robustness verification were performed by numerical simulations. The logic of this manuscript is clear, the figures are of high quality and the results are convincing. I recommend it to publish after addressing the following issues:
1) Some font formatting errors need to be noted, such as “Fig. 3 gives the dispersion relation of unit cells with different Ld” in line 155 should be “Fig. 3 gives the dispersion relation of unit cells with different Ld”.
2) in Figure 3, the authors say that the upper bandgap is locally resonant bandgap. The question is which one is the resonance branch? The upper one? What is the mode of resonance in particular? I think the discussion here needs to be expanded.
3) in Figure 6, The acquisition process of the transmittance curve is not detailed enough. Where is the displacement detection point? Where is the excitation point? What is the algorithm for the transmittance?
4) in Figure 9(a), the curve peak near 565Hz also seems to have good robustness, and there is no topological interface state here. Can the authors comment on the robust mechanism?
5)Some recent publications on topological metamaterial beams are highly related with this manuscript such as[Physical Review B 101 (2), 024101,2020], [International Journal of Mechanical Sciences 186, 105897,2020]&[Science China Physics, Mechanics & Astronomy 65 (1), 214612,2022] which can be considered.
Author Response
Please see the attachment. Thanks so much for your help to the quality of the manuscript.

Reviewer 2 Report
Comments on the manuscript
In this manuscript entitled “Vibration energy harvesting from the subwavelength interface state of a topological metamaterial beam”, the authors introduced the simulation results of the realization of a topological metamaterial for vibration energy harvesting. They presented several interesting results, including energy gathering at the interface by the involvement of defects in the vicinity of the conjunction. The idea in this manuscript, in general, is interesting, and the simulation results are incremental to the field of topological metamaterials. The manuscript is technically sound with conclusions and assertions well-supported by simulation results. However, this manuscript did not provide deep insight into physics but instead presented superficial results. Also, the language is a little difficult to understand with flaws in accuracy and fluency. These two factors together degrade the whole level of the manuscript.
Thus, this manuscript needs substantial revision before it can meet the scope of Micromachine. Consequently, there are a few points that should be addressed for the purpose of clarity.
- My major concern is the scientific insight of this manuscript. The authors presented interesting simulation results with sharp resonances at the conjunction interface, which is nice and attractive to the readers. However, the physical links between the topological surface states and the involved defects were not presented. This would lead to some potential problems for readers who focus on the physical theory aspects. I would recommend that more explanation is required. The author needs to think seriously about this issue since it is closely related to the technical innovations and scientific impact of the manuscript.
- Also, in figure 9, it appears that the defect may have a profound impact on the resonance intensity around it, why is that a hot spot occurs at the first unit cell whereas the defect is at the 9th unit cell (figure 9b)? please comments
- Have the authors considered the finite-size effects? Their structures only have 5 elements on both sides.
Author Response

(The authors gave the same response as above.)

Round 2
Reviewer 2 Report
The authors have addressed my comments and improved the manuscript substantially. The manuscript is scientifically sound. Thus, I have no further questions but to give my proposal of acceptance.